# Validation of COSMIC water vapor data in the upper troposphere and lower stratosphere using MLS, MERRA and ERA-Interim

Ming Shangguan<sup>1</sup>, Katja Matthes<sup>1,2</sup>, Wuke Wang<sup>1</sup>, and Tae-Kwon Wee<sup>3</sup>

<sup>1</sup>GEOMAR Helmholtz Centre for Ocean Research Kiel, Kiel, Germany

<sup>2</sup>Christian-Albrechts-Universität zu Kiel, Kiel, Germany

<sup>3</sup>University Corporation for Atmospheric Research, Boulder Colorado, USA

Correspondence to: Ming Shangguan (mshangguan@geomar.de)

**Abstract.** Water vapor is the most important greenhouse gas in the atmosphere with important implications not only for the Earth's radiation and energy budget but also for various chemical, physical and dynamical processes in the stratosphere. The Constellation Observing System for Meteorology, Ionosphere and Climate (COSMIC) Radio Occultation (RO) dataset from 2007 through 2013 is used for the first time to study the distribution and variability water vapor in the upper troposphere

- and lower stratosphere (UTLS). The COSMIC data are compared to the Microwave Limb Sounder (MLS) data, and to two global reanalyses: The Modern-Era Retrospective analysis for Research and Application (MERRA) of the National Aeronautics and Space Administration (NASA); and, the latest reanalysis of the European Center for Medium-range Weather Forecast (ECMWF), the ERA-Interim. The MLS data have been assimilated into the MERRA, whereas the COSMIC data are used for the ERA-Interim. As a result, the MERRA agrees well with the MLS data and so does the ERA-Interim with the COSMIC
- data. While the monthly zonal mean distributions of water vapor from the four datasets show good agreements in northern mid-latitudes, large discrepancies exist in high southern latitudes and tropics. The MERRA shows overall a consistent seasonal cycle with MLS, but has too strong winter dehydration over the Antarctic, and is very weak in the interannual variations. The ERA-Interim fails to properly represent the winter dehydration over the Antarctic, and shows an unrealistic seasonal cycle in the tropical upper troposphere. The COSMIC data shows a good agreement with the MLS data except for the tropical "taper
- recorder" signal, where the COSMIC data suggest a faster upward motion than the MLS data. The COSMIC data are able to represent the moisture variabilities associated with the Quasi-Biennial Oscillation and the El Niño-Southern Oscillation.

#### 1 Introduction

Water vapor plays an important role in the Earth's atmosphere by transporting energy and involving in various atmospheric processes. Water vapor is the dominant greenhouse gas, introducing effective long-wave radiative forcing in the atmosphere

(Forster and Shine, 2002). As such, an increase of water vapor warms the surface, which in turn boosts the evaporation, supplying more moisture to the atmosphere. The cycle relates to a strong positive feedback. The radiative absorption by water vapor is roughly proportional to the logarithm of its concentration. This means that the influence of an increase in water vapor is more prominent in regions where the moisture concentration is relatively low, such as in the upper troposphere and lower stratosphere (UTLS) (Change Climate, 2007, 2013). Previous studies reported significant variations of water vapor in

the UTLS, including: the sudden drop of water vapor around 2001 (Urban et al., 2014); interannual to decadal variability (Schieferdecker et al., 2015); and, long-term trends (SPARC, 2000; Rosenlof et al., 2001; Hurst et al., 2011). Forster and Shine (1999, 2002) emphasized that an increase of water vapor can have a significant contribution to the cooling of the lower stratosphere and a warming of the troposphere. Even small water vapor changes in the lower stratosphere can result in strong

- 5 radiative effects on surface climate (Solomon et al., 2010). Therefore, a better knowledge of water vapor contents in the UTLS is important and urgently required for understanding surface climate and the thermal structure of the UTLS (Forster and Shine, 1999; SPARC, 2000). The UTLS water vapor is of great interest, for example, the SPARC reanalysis intercomparison (S-RIP) project (s-rip.eeshokudai.ac.jp/report/structure) focuses on the climatology and interannual variability of water vapor as well as the tropical tropopause layer. The water vapor in turn impacts the UTLS thermal structure through the radiative transfer. For
- 10 example, it plays a major role in maintaining the temperature inversion and the tropopause inversion layer (TIL) structure (e.g., Randel et al., 2007; Kunz et al., 2009; Hegglin et al., 2010; Randel and Wu, 2010). On the other hand, because of its strong vertical gradient from the troposphere to the stratosphere, it can be used as a tracer to study stratosphere-troposphere exchange processes and atmospheric dynamics (Pan et al., 2007).
- Information on water vapor can be obtained from remote sensing measurements. A variety of satellite instruments, e.g.,
  MLS (Microwave Limb Sounder), SCIAMACHY (Scanning Imaging Absorption Spectrometer for Atmospheric Cartography),
  MIPAS (Michelson Interferometer for Passive Atmospheric Sounding), are used to detect water vapor (Schoeberl et al., 2006;
  Rozanov et al., 2011; Fischer et al., 2008). Besides the satellite data, water vapor can be also measured by aircraft, radiosonde and ground-based instruments (radiometer, lidar, spectrometer etc.) (Kiemle et al., 2008; Sitnikov et al., 2007; Miloshevich et
- al., 2009). However, measuring water vapor in the UTLS is challenging because of the low stratospheric mixing ratios, large
  moisture gradients around the tropopause, and complicated radiative effects due to clouds (SPARC, 2000). Further, most of the above-mentioned satellite measurements are coarse in the vertical resolution, whereas the ground-based observations are poor in the horizontal coverage, which are both important for an adequate description of the moisture distribution in the UTLS. Model simulations, reanalysis datasets, and satellite data in the region show discrepancies among themselves in the distribution of water vapor (Solomon et al., 2010; Vergados et al., 2015).
- Since 2001, the Global Positioning System (GPS) Radio Occultation (RO) technique has been providing wealthy data for monitoring the global atmosphere. The RO measures the time delay in occulted signal using the GPS receiver on board a low Earth orbiting (LEO) satellite, which can be transformed to atmospheric bending angle and refractivity, and then again to profiles of atmospheric pressure, temperature (Wickert et al., 2001), and water vapor (Wickert et al., 2004). Possessing unique aptitudes of high accuracy, weather independence, global coverage, and self consistency, the RO temperature has been
- widely used in the UTLS region (e.g., Grise et al., 2010; Schmidt et al., 2010; Wang et al., 2013; Randel, 2015). In contrast, RO-retrieved water vapor data has been less utilized.

Several studies (Heise et al., 2006; Ho et al., 2010; Sun et al., 2010) have compared the RO water vapor data with reanalyses, radiosondes and other satellite observations. Kishore et al. (2011) found that RO water vapor profiles have high precision up to 7-8 km from the surface in tropical and mid-latitudes by comparing the RO water vapor from the Constellation Observing System for Metaorelogy Japageberg and Climate (COSMIC) mission with CBS radiosonde observations and the reanalyses.

System for Meteorology, Ionosphere and Climate (COSMIC) mission with GPS radiosonde observations and the reanalyses.

5

Wang et al. (2013) concluded that the COSMIC water vapor could be used as a benchmark to evaluate other observational data based on the comparison with radiosonde data from 925 hPa to 200 hPa in the period from 2007 to 2010. Vergados et al. (2015) compared COSMIC water vapor observations with the Modern-Era Retrospective analysis for Research and Application (MERRA) data of the National Aeronautics and Space Administration (NASA) in the lower and middle troposphere from 2007 to 2009. They found a good agreement between the MERRA reanalysis and the COSMIC data in the tropical region. However, neither of these comparison studies is focused on the water vapor in the UTLS region. We therefore propose a first validation of the COSMIC water vapor in the region, including the spatial distribution, and seasonal to interannual temporal variations. To do so, we take an independent source of observation, the MLS, as a reference and also make comparisons to the ERA-Interim and MERRA reanalyses.

#### 10 2 Data and comparison methods

We analyze the distribution and variability of water vapor using COSMIC, ERA-Interim, MERRA and MLS datasets during 2007-2013. First, we generated monthly zonal mean time series of the volume mixing ration (VMR) on pressure levels for which the COSMIC data are converted from the water vapor pressure. These time series serve as the basis for our study. The validation includes comparisons of monthly and annual zonal mean averages over the seven-year period. In addition, the

- 15 interannual variability in the datasets are analyzed with time series of deseasonalized anomalies for selected latitude bands and pressure levels. The monthly anomaly for each month is calculated by subtracting the long-term monthly mean from individual monthly mean. ENSO and QBO indices are also used for the analysis. The Multivariate ENSO index (MEI) time series since 1950 to present is available at www.esrl.noaa.gov/psd/enso/mei/. The QBO index at 50 hPa from 1953 to 2015 can be found at www.geo.fu-berlin.de/en/met/ag/strat/produkte/qbo.
- It has to be mentioned that the datasets differ in the spatial and temporal resolutions. In particular, the observation density of COSMIC and MLS over a relatively small area varies time to time depending on the geometry and condition of the observing systems introducing an additional uncertainty to the monthly means. Although the reanalyses are produced combining observations from a number of different data sources, the number of high-quality observations could still be often insufficient especially in tropical and polar regions. The ERA-Interim used in this study is with 13 pressure levels, coarser than other
- 25 datasets: 21 for MERRA; 19 for MLS; and, 55 for COSMIC. All the datasets were interpolated to the same pressure levels as MLS whereas necessary, i.e., Figures 4-6, for the comparison. The different vertical resolutions do not influence the general feature because sharp features like the hygropause are smeared out by averaging over a month due to tropopause height fluctuations.

#### 2.1 Satellite data sets

## 2.1.1 COSMIC

GPS RO is a relatively new remote sounding technique for monitoring the global atmosphere: The receiver on board a low Earth orbiting (LEO) satellite records occulted GPS signals; the time-frequency contents in the received signals, with the knowledge

- of high-precision orbits of the LEO and occulting GPS satellites, can deduce a profile of atmospheric bending angle; the inverse 5 Abel transform can then yield a refractivity profile from the bending angle profile (Fjeldbo et al., 1971). Finally, profiles of atmospheric pressure, temperature and water vapor in the UTLS can be estimated from the refractivity profile (Kursinski et al., 2000; Heise et al., 2006). Among a few GPS RO missions, COSMIC has been providing the majority of data since its launch in 2006, about 2000 profiles daily, on average, across the globe during our study period.
- We use the COSMIC data (wetPrf) provided by the COSMIC Data Analysis and Archive Center (CDAAC) between 2007 and 2013 in this study. The CDAAC reprocesses past RO data with the intent to provide consistent, high-quality data sets for climate studies, in addition to providing near real-time data for operational weather forecasting (http://cdaac-www.cosmic.ucar.edu). The COSMIC data used in this study are post-processed wetPrf version 2010.2640. These profiles have a 100 m vertical resolution between 0-40 km, and are produced by a one-dimensional variational retrieval method (1D-Var). Wee and Kuo
- (2014) describe the variational framework on which the CDAAC 1D-Var is built. The 1D-Var seeks for the optimal solution that reproduces the observed refractivity most closely while satisfying other imposed constraints, amongst many physically admissible combinations of state variables (i.e., non-negative and sub-saturated water vapor sub-adiabatic temperature and hydrostatic pressure).

The amount of water vapor in the atmosphere varies substantially with time and space, and is thus difficult to measure. This is especially true in the UTLS, where water vapor varies by a few orders of magnitude. The lack of reliable moisture 20 observations in the region also hampers a precise error characterization of the background water vapor. Like other variational approaches, error covariance matrices for both, observation and background, play a crucial role in the 1D-Var. The extent of correction to be applied to the background moisture differs depending on the assumed accuracy of background water vapor compared to those of temperature and pressure. The inter-variable relative accuracy is weighed by converting the background error of all parameters into the common refractivity unit.

As shown by Wee and Kuo (2014), a variational approach is expected to yield retrievals of a rigorous accuracy. Even so, it is not straightforward to validate the quality of RO-retrieved water vapor in the stratosphere because the amount of moisture is so small that it may challenge the precision and accuracy of any verifying observations. In addition, the 1D-Var does not account for profile-to-profile variations in the background water vapor error. On a profile-by-profile basis and in weather perspectives,

it is therefore difficult to assess the quality of COSMIC water vapor in the UTLS unless the moisture reaches a level clearly 30 detectable by other verifying observations, e.g., when the lower stratosphere becomes saturated in the events of overshooting convection or polar stratospheric clouds.

As an alternative, we propose to validate the COSMIC water vapor in a climate perspective. RO phase measurements are known to be very precise and accurate (e.g., Wee and Kuo, 2015), leading to high-quality data for climate studies (e.g., Ho et

al., 2012; Steiner et al., 2013). It is also shown that the variational method can produce profiles of refractivity, temperature, and pressure that agree better with radiosonde observations than operational ECMWF analyses (Wee and Kuo, 2014), although the radiosonde data are assimilated into the ECMWF analysis system. The studies also confirm that RO-derived profiles are virtually bias-free, and small systematic differences from radiosonde data are largely attributable to radiation-induced biases in radiosonde temperatures (Sun et al., 2013). Given that the measurements are accurate and all other derived parameters are unbiased, it would be reasonable to expect the derived water vapor to be accurate as well.

The background error covariance matrix used in COSMIC 1D-Var is based on long-term statistics of background departures from radiosonde observations. The differences between short- and longer-term forecasts valid at the same are also used to extend the coverage above the balloon burst height. In regard to the moisture in the UTLS, the background error is about 10%

- on average in the relative humidity and it varies with height, latitude, and season. The error correlation length used in the 1D-Var is about 200 m in that region. In this study, we hypothesize that the variational estimation is robust for the large-scale. In other words, we anticipate the method to produce a realistic distribution of water vapor in the large spatiotemporal scales for which the assumption we made about the background error is more likely to hold. By averaging a large number of samples collected over a long period, occasional retrieval errors in individual profiles may cancel each other out. The massive sample
- size also reduces the uncertainty in the mean water vapor. In comparing the COSMIC water vapor to other datasets, our focus is on the scales that can be commonly represented by all datasets. Therefore, we converted the water vapor pressure into VMR and then interpolated it to 55 fixed pressure levels between 10 hPa and 300 hPa. The horizontal grid used in this study for the COSMIC data is a regular 1.9°x2.5° latitude-longitude grid.

# 2.1.2 MLS

- MLS was launched into a near polar, sun-synchronous, 705 km altitude orbit on 15 July 2004 (Schoeberl et al., 2006). It scans the Earth limb giving 240 scans per orbit, spaced about 165 km along the orbit track, and up to 3500 profiles per day. It has a global latitudinal data coverage from  $82^{\circ}N$  to  $82^{\circ}S$ . MLS measures the radiance emitted by the Earth's atmosphere and water vapor is retrieved from measurements of the 190 GHz water vapor rotational line spectrum. The MLS data processing algorithm is based on the optimal estimation approach and uses a two-dimensional system to determine temperature, geopotential height
- and composition (Livesey et al., 2006). Here, we use the latest public release of MLS data v4.2, in which the upper tropospheric and lower stratospheric humidity estimation and cloud detection methodology are improved compared with earlier MLS data versions. Especially the dry biases of MLS data at 316-215 hPa is better than earlier version though there are still some extremely low values at 215 hPa (Livesey et al., 2015). The MLS measurements from 2007 to 2013 were compiled into monthly zonal mean time series on a 1.5°x2° latitude-longitude grid, and on 20 pressure levels from 316 hPa to 10 hPa. In
- April 2007, 2008, 2009, 2010 and 2011 and November 2007, 2008, 2009, 2010 and 2013 there is no valid water vapor data in the MLS data.

The MLS water vapor data has been validated by a number of studies with different instruments, such as datasets from balloon and satellite platforms, frost point hygrometer and WB57 aircraft hygrometer (Lambert et al., 2007; Read et al., 2007; Hurst et al., 2014). The MLS water vapor shows good agreement with the multi-instrument mean reference throughout most of

the atmosphere (Hegglin et al., 2013). Details about the accuracy, precision and quality of MLS water vapor data are described in Livesey et al. (2015). Therefore, we use the MLS data as a reference to evaluate the water vapor volume mixing ratios derived from COSMIC as well as the MERRA and ERA-Interim datasets.

# 2.2 Reanalysis Datasets

# 5 2.2.1 ERA-Interim

ERA-Interim is one of the most advanced global atmospheric reanalyses representing the state of the atmosphere using a 4D-Var method (Simmons and Hollingsworth, 2002). The 4D-Var assimilates a number of different sources of observation such as radiosonde humidity, AIRS (Atmospheric Infrared Sounder) radiance, GPS RO bending angle profiles, SSM/I (Special Sensor Microwave/Imager), ERS (European Remote Sensing Satellite)-1 and -2 etc. (Dee et al., 2011). Here we use monthly ERA-

10 Interim data with a horizontal resolution of 1.5°x1.5° on 37 pressure levels between 1 hPa and 1000 hPa. In this study, we only use 13 levels from 300 hPa to 10 hPa. The specific humidity in the reanalysis data is converted into VMR.

## 2.2.2 MERRA

MERRA is a reanalysis dataset based on the Goddard Earth Observing System Data Assimilation System Version 5 (GEOS-5) (Rienecker et al., 2008). MERRA takes advantage of a variety of recent satellite data, for example the NASA's Earth

15 Observing System (EOS), the AIRS instruments, the Advanced Television and Infrared Observatory Spacecraft Operational Vertical Sounder (ATOVS) etc. (Rienecker et al., 2011). We use monthly MERRA data on 1.5°x2° latitude-longitude grid and 21 levels. The original data have 72 levels that are about 1 km in the resolution. While using MLS data, MERRA does not assimilate COSMIC data. Therefore, COSMIC is independent of MERRA and MLS is unrelated to ERA-Interim.

#### 3 Climatological Distribution and Variability of Water Vapor

#### 20 3.1 Climatologies

Figure 1 compares the pressure-latitude distribution of water vapor averaged over the entire period from 2007 to 2013, where the MLS is shown above the MERRA and the COSMIC above the ERA data. The panels are laid in the way considering the fact that MERRA assimilates the MLS and ERA-Interim does COSMIC.

In general, water vapor increases rapidly with an increasing pressure in the upper troposphere, and tropics are wetter than high latitudes reflecting the meridional thermal structure. The tropopause height, which varies with the latitude, can be roughly estimated from the sudden changes in the vertical moisture gradient there. Above the tropical tropopause (near 80 hPa), the minimum VMR is ~3.5-4 ppmv. This can be explained by dehydration of water vapor while crossing the cold tropopause as reported by previous studies (e.g., SPARC, 2000; Schoeberl et al., 2012; Hegglin et al., 2013). At higher levels, the water vapor increases slightly with the height, related to oxidation of methane (e.g., Mote et al., 1996; SPARC, 2000). Both COSMIC and

30 MLS capture all these features well, although there are some differences between them in the detailed pattern and amount. The

two reanalyses are also reasonable in the broad feature. However, ERA-Interim shows weaker minima in the tropics and over the high southern latitudes around 100 hPa, compared to other data.

Figures 2 and 3 show the water vapor distribution during boreal winter (December, January, February) (DJF) and summer (June, July, August) (JJA). The dryness around the tropical and Antarctic tropopauses is marked during DJF and JJA,

- respectively. The inter-data difference is largest around these two minima of VMR. For DJF (Figure 2), COSMIC agrees well 5 with MLS, although it is slightly wetter than MLS around the tropical troppoause and drier in the tropical lower to middle stratosphere (50-10 hPa). MERRA also shows a good agreement with MLS in the tropics but is too wet over the Antarctic, ERA-Interim on the other hand is persistently wetter in both regions. For JJA (Figure 3), the inter-data discrepancy is more pronounced over the Antarctic. While all datasets show lowest VMR from lower to middle stratosphere resulting from very low
- temperatures and subsequent dehydration, ERA-Interim is noticeably wetter than other datasets over the Antarctic. MERRA 10 captures this minimum, but with another local minimum below the tropopause over latitude  $60^{\circ}S$  to  $80^{\circ}S$  between 150 hPa and 300 hPa, which is not supported by COSMIC observations. The disagreement among the datasets over the Antarctic is presumably due to the limited data coverage of MLS in the latitudes southward of  $82^{\circ}S$ . COSMIC, which does not have such an issue, might be trustworthy in the region. Nonetheless, the small VMR (~2 ppmv) may challenge the accuracy of any
- 15 present-day stratospheric water vapor measurements.

To further understand the inter-data differences between these datasets, percentage differences of annually averaged water vapor profiles (300-10 hPa) are shown in Figures 4 for five latitudinal bands: tropical  $(10^{\circ}N-10^{\circ}S)$ , mid-latitudes  $(45^{\circ}N-10^{\circ}S)$  $60^{\circ}N$  and  $45^{\circ}S$ - $60^{\circ}S$ ) and polar ( $70^{\circ}N$ -  $80^{\circ}N$  and  $70^{\circ}S$ - $80^{\circ}S$ ). We averaged all data that belong to each band for each dataset at different pressure levels and then interpolated to the MLS pressure levels in order to use the MLS data as a reference

20 (Figure 4-6).

> The most evident discrepancies of water vapor between the different datasets are found in the upper troposphere region (250-150 hPa), which can be seen in all latitude bands. COSMIC shows 50-100% higher water vapor than MLS in the tropics and over 100% higher water vapor at other latitudes. ERA-Interim shows comparable or even larger discrepancies than COSMIC. MERRA shows the best agreement with MLS but is also biased somewhat higher, although it assimilates MLS data. This is

- because of the known dry (low) bias of MLS water vapor retrievals in the UTLS region (Vömel et al., 2007; Barnes et al., 25 2008; Hegglin et al., 2013; Hurst et al., 2014). Figure 4 also demonstrates that, COSMIC and MERRA are about 10% lower than MLS and ERA-Interim about 20% at 50-10 hPa over the Antarctic. In the southern mid-latitudes at 50-10 hPa MERRA is similar with MLS, ERA-Interim is about 20% lower than MLS and COSMIC is between the two about 10% lower than MLS. In the tropics, COSMIC and ERA-Interim show higher values (about 10%) from 100 to 50 hPa and lower values (about
- 5%) from 30 to 10 hPa compared to MLS. In the Southern Hemisphere (SH) mid-latitudes COSMIC and ERA-Interim show 30 consistent larger values (about 20%) than MLS in the stratosphere. In the Northern Hemisphere (NH) middle to high latitudes, all datasets are in very good agreement in the stratosphere ( $\sim 10\%$  less than MLS).

Zonal band profiles averaged in DJF and JJA are further presented in figure 5-6. Discrepancies between COSMIC, ERA-Interim, MERRA and MLS are similar to those in figure 4. Exceptions can be found in the Antarctic in JJA, where ERA-Interim values are too high ( $\sim$ 35%) from 70 to 30 hPa. This is because of the supersaturation problem in ERA-Interim (Dee et al.,

35

5

2011), which will be discussed in more detail later. Another exception are significant differences between the other data and MLS in the tropical stratosphere. COSMIC, ERA-interim and MERRA are higher ( $\sim$ 30% by COSMIC,  $\sim$ 20% by ERA-Interim, and  $\sim$ 5% by MERRA) from 70 to 50 hPa but lower ( $\sim$ 10%) from 50 to 30 hPa than MLS in JJA. In contrast, there are lower values ( $\sim$ 10%) from 70 to 50 hPa and higher values ( $\sim$ 15% by COSMIC,  $\sim$ 12% by ERA-Interim, and  $\sim$ 5% by MERRA) from 50 to 30 hPa in DJF, which cannot be observed in the annual mean.

In summary, for the water vapor climatologies, MERRA shows the best agreement with MLS in general, since it assimilates the MLS. COSMIC is consistent with MLS in the middle stratosphere, but is wetter than MLS. Considering the dry bias of MLS in the UTLS, COSMIC has a good quality in water vapor from the upper troposphere to the middle stratosphere, and is very valuable for studying the UTLS since it has much higher vertical resolution (100 m) than satellite instruments like MLS (2

10 km or more) and reanalysis data (more than 1 km). ERA-Interim is close to the COSMIC data, since it assimilates this dataset but has a pronounced wet bias over the Antarctic latitudes.

#### 3.2 Seasonal Variability

The seasonal cycle of water vapor at different pressure levels, including its annual cycle and interannual variations, from 2007 to 2013 averaged within the chosen latitudinal belts ( $10^{\circ}N$  to  $10^{\circ}S$ ,  $45^{\circ}N$  to  $60^{\circ}N$ ,  $45^{\circ}S$  to  $60^{\circ}S$ ,  $70^{\circ}N$  to  $90^{\circ}N$ ,  $70^{\circ}S$  to

15  $90^{\circ}S$ ) are shown in Figures 7-9. Note that the MLS data are only available until 82°. Water vapor exhibits a strong seasonal cycle, which is the main source of variability for water vapor in the UTLS region. The tropical region between  $10^{\circ}N$  and  $10^{\circ}S$  was chosen in order to get a strong signal.

Figures 7-9 show the annual evolution of UTLS water vapor, which has been averaged for the period from 2007 to 2013 in the same latitude bands as above, as a function of month and pressure.

- The Antarctic region exhibits a strong seasonal cycle in the stratosphere, with lowest values in winter and highest values in summer (Figure 8). The lowest values are due to the strong dehydration of water vapor during very cold winters (Kelly et al., 1989). COSMIC and MLS observations are consistent with each other (Figures 7a and c). The reasons for the differences between different datasets could be caused by the uncertainty of the measurements as well as intermittent sampling. MERRA data show a general agreement with MLS, although the dehydration in winter reaches too deep into the troposphere, which
- 25 results in very low values of water vapor between 200 and 150 hPa. The ERA-Interim is very different from other datasets in the seasonal variability (Figure 7d), not showing the winter dehydration. According to the description in Dee et al. (2011), the relative humidity in the stratosphere has increased over the southern polar cap as a result of the introduction of supersaturation in the assimilation algorithm of the new model version compared to ERA-40. It explains the obviously high water vapor values of ERA-Interim in this region compared to all other datasets.
- 30 The Arctic region shows a different annual cycle than the Antarctic (Figure 8). During boreal winter, since the polar vortex in the NH is not as steady and strong as in the SH, the temperature is not low enough for water vapor dehydration. At the same time, the downward motion of the Brewer-Dobson circulation (BDC) is strongest during winter, which transports wet air from the upper levels to the lower stratosphere. Therefore, the minimum water vapor is not in winter, but exists during spring to summer instead. Also interesting is that the seasonal cycle is different in the upper troposphere compared to the lower

stratosphere. This is because the water vapor in the upper troposphere is controlled by the tropopause height and the mixing in the lower atmosphere. The maximum is shifted to summer, because a higher tropopause allows for more tropospheric air mixing. The upper tropospheric water vapor is effected by the surface and the stratospheric water vapor is more effected by the upper stratosphere, which can be clearly seen from the slopes of the contour lines (especially for COSMIC). The annual cycle in the reanalysis datasets agrees well with observations in general, although the amplitudes are weaker in both MERRA and

5

ERA-Interim than in MLS and COSMIC.

In mid-latitudes (not shown here), the strong seasonal variability in the upper troposphere in both hemispheres is similar to those in the NH polar region, i.e., the bottom-up transport determines the maximum of water vapor in summer at the upper troposphere and the top-down transport determines the minimum water vapor in spring to summer in the lower stratosphere.

- Unlike the similar value in the upper troposphere in the NH, the ERA-Interim is much higher and MERRA is much lower than 10 MLS and COSMIC data in the upper troposphere (not show here). The SH lower stratosphere between the tropopause (around 100 hPa) is slightly drier than the NH, partly due to moistening by the northern mid-latitudes monsoon (e.g., Stone et al., 2000; Randel, 2015). The ERA-Interim in mid-latitude agrees better with satellite data (COSMIC and MLS) than MERRA.
- In the tropics, water vapor is lowest during boreal winter (DJF), when the tropopause is coldest and water vapor is mostly 15 dehydrated (Figure 9). The observed seasonal variation of water vapor at the tropical tropopause is affected by the annual variation in tropical tropopause temperatures (SPARC, 2000). This very dry air is then transported upwards to about 10 hPa, which is termed as the tropical "tape recorder" (Mote et al., 1996). The slope of this "tape recorder" indicates the speed of the upwelling. The upwelling is faster in COSMIC and ERA-Interim compared to MLS and MERRA. The faster tropical upwelling in ERA-Interim than in MERRA has been already noticed in previous studies (e.g., Schoeberl et al., 2012). COSMIC
- water vapor retrievals are based on an ERA-Interim temperature background, which might be the reason for this faster tropical 20 upwelling in COSMIC. Also note that the satellite sampling biases have a larger impact on the annual mean around the tropical tropopause, where the natural variability is large (Toohey et al., 2013). MERRA shows a slower and shallower "taper recorder" signal, which indicates a slower tropical upwelling in MERRA. In the upper troposphere the seasonal cycle of water vapor is relatively weak (Newell et al., 1997), however, ERA-interim shows an unrealistic annual cycle in the upper troposphere, which
- cannot be seen in other datasets. 25

In summary, for the seasonal variability of water vapor, different regions produce different features since they are dominated by different processes mentioned above. COSMIC shows good agreement with MLS in general, except for the about 50% faster tropical "tape recorder" in the tropics. MERRA shows a correct seasonal cycle in general, but has a too strong winter dehydration in the Antarctic region. ERA-Interim does not have or has only a very weak winter dehydration in the Antarctic region and produces an unrealistic seasonal cycle in tropical upper tropospheric water vapor.

30

#### 3.3 **Interannual Variability**

In addition to the seasonal variations described above there are also interannual variations in water vapor. Interannual variations are related to many different effects such as the Quasi-Biennial Oscillation (QBO) in the stratosphere, ocean-atmosphere interactions such as El Niño-Southern Oscillation (ENSO), and polar vortex temperatures. The evaluation of interannual vari-