# Peer review of "Validation of COSMIC water vapor data in the upper troposphere and lower stratosphere using MLS, MERRA and ERA-Interim"

_Atmospheric Measurement Techniques, 2016_

## Referee Comment (RC1) · Anonymous Referee #1 · 23 Aug 2016

Review of "Validation of COSMIC water vapor data in the upper troposphere and lower stratosphere using MLS, MERRA and ERA-Interim" by M. Shangguan et al. (amt-2016-248)

COSMIC radio occultation data form a valuable contribution to the observation system of the atmosphere. They provide, according to their website http://www.cosmic.ucar.edu/ro.html, temperature up to the stratopause and water vapour abundances up to about 10 km altitude, both with dense coverage and very high vertical resolution. The manuscript aims to provide validation of COSMIC water vapour data up to about 30 km altitude (10 hPa). Should COSMIC water vapour data

indeed be useful in the stratosphere as well, this would be an important finding and an immense asset for atmospheric observations.

According to the title, the paper claims to provide a validation of upper tropospheric and lower stratospheric water vapour derived from COSMIC radio occultation measurements. The COSMIC data are compared within an approach explicitly not based on collocations (but on basis of zonal monthly or even multi-annual means) to MLS satellite data and assimilated products as MERRA and ERA-I. The authors argue that such a validation is appropriate in a climate perspective (p4, l33). It is pointed out that COSMIC data are assimilated into ERA-I so that is not an independent data set, while MERRA is free from COSMIC data but linked to MLS as MLS data is assimilated into MERRA. It is concluded that COSMIC agrees well with MLS except for the tropical tape recorder for which COSMIC data suggest a faster upward motion. QBO and ENSO related signatures are also contained in COSMIC data.

There are several fundamental problems with this paper:

1. One main problem with remote sensing data in general is the content of information from the measurement versus *a priori* information in the profiles provided. The COSMIC water vapour data are reported on the COSMIC web site (http://www.cosmic.ucar.edu/data.html) to be valid and useful for the altitude range below 10 km. In this manuscript, the water vapour data from COSMIC are used up to 10 hPa ( 30 km). No justification is given why this extension of the valid altitude range is reasonable, and no assessment is given on the information content of COSMIC data above its recommended altitude range. I am very sceptical about any information on stratospheric water vapour contained in the COSMIC radio occultation data because of the following simple back-of-the envelope assessment: According to Ho et al. (2010), their Eq. (1), the refractivity of the atmosphere is given by:

N = 77.6 p/T + 3.73E5 $p_W/T^2$

Evaluating this formula with typical stratospheric values of $p_W$ = 5E-6 * p ( 5 ppmv volume mixing ratio) and T = 250 K gives

N = p/T (77.6 + 0.0075)

i.e. the impact of typical stratospheric water vapour mixing ratios on the refractivity is in the order of 1E-4 (0.01%). In other words, the refractivity must be measured with an uncertainty better than 0.01% to derive information on water vapour from the radio occultation measurements. The authors need to provide evidence that this high precision is achieved with the COSMIC measurements.

Assuming that there is no or only marginal information on stratospheric water vapour in the data from COSMIC measurements, the 1D-var retrieval as described in the document "Variational Atmospheric Retrieval Scheme (VARS) for GPS Radio Occultation Data" (http://cdaac-www.cosmic.ucar.edu/cdaac/doc/documents/1dvar.pdf) would just reproduce the *a priori*. There is no information easily accessible what the *a priori* ($\mathbf{x}_b$ in the terminology of the CDAAC document) is, and the authors do not mention it either. However, Ho et al. (2010) state that data from the NCEP Global Forecast System model are used as first guess. Assuming that first guess and *a priori* are identical (a usual practice in satellite data retrievals), this would mean that COSMIC retrievals based on measurements without information content on stratospheric water vapour would just reproduce the NCEP fields. If so, the presented study would have no validation power.

NCEP, as a meteorology product, must be extended by any reasonable climatology into the stratosphere (I do not know which one). Therefore, it must be shown that

the COSMIC measurements contain independent information on stratospheric water vapour; this could be done by analysing the averaging kernels. In other words, it remains to be shown that the COSMIC stratospheric water vapour fields are different from the *a priori* information used in the retrievals. This can be done by comparing the *a priori* fields used in the retrievals with the retrieved water vapour fields.

2. The paper does not present a validation in the sense this term is used within the satellite community. It compares monthly up to multi-annual zonal means of COSMIC data to one other satellite data set (MLS) and two assimilated products, one of which is dependent on COSMIC (ERA-I), and the other is linked to MERRA. I.e. in fact there is one single independent data set (MLS) the COSMIC data are compared to. The comparison is not made on basis of collocations, but for monthly or even multi-annual zonal means; from this type of comparison it can be derived at best if the general climatological features of UTLS water vapour are reproduced by COSMIC water vapour. Indeed the authors state that this was their aim, and the discussion of the comparisons is restricted to these topics (e.g. de-hydration in the Antarctic polar vortex, signature of the tape recorder). **Any halfway reasonable climatology would pass this kind of validation.** Differences are shown for multi-annual means only and are not discussed. There is no way by this approach to quantify any situation-dependent biases or detect situations/processes where COSMIC is exceptional good or bad. Again the question arises if not simply the *a priori* climatology has been validated.

3. From Figs. 4 - 6 one must conclude that the tropospheric bias of COSMIC wrt MLS is mostly positive and often 50 to 100% or even more. However, this is the altitude validity range of COSMIC according to its web site; MLS, on the other hand, is known to be biased slightly low in this altitude range (but not more than 10%). Is this result (COSMIC wet bias of 40 to more than 100% in the troposphere) consistent to earlier validations cited in this paper? These earlier validations obviously stated that COSMIC

is in good agreement and "virtually bias-free" wrt radiosonde data (page 5, line 4)? In my opinion, the results of this study are not put properly into context of earlier validation studies, and obvious discrepancies are not discussed.

There are a number of minor issues I have noticed but I am not going to comment on these in detail:

- Figs. 1 – 3 should be presented as differences to MLS in order to provide useful information; from the Figures as presented now, it is very hard to derive quantitative information.

- When comparing zonal means care has to be taken that the representativeness due to different sampling does not produce a bias (see e.g. Toohey et al., JGR, 2013).

- Given that the title refers to validation of COSMIC data, the evaluation of MERRA and ERA-I assimilation products is off focus and is more confusing than helpful here. If the authors wish to keep this focus, the title of the paper (and the introductory remarks) should be adjusted accordingly.

In summary, the authors have not provided sufficient evidence that COSMIC indeed provides information on stratospheric water vapour. It has not been shown that the validation performed here has not just validated the *a priori* information used in the retrievals. The validation approach is not appropriate to judge if the COSMIC data indeed contain any information on stratospheric water vapour.

A proper validation approach

- would present and discuss differences and quantify biases in relation with their uncertainties

- would discuss which conclusions are possible from the validation and which are not, i.e. would discuss the limits of the validation

- would put its result in context to earlier validation studies.

Given these concerns, I cannot recommend this paper for publication. I am afraid that once published, it would be used and cited as justification for using COSMIC water vapour data in the stratosphere. I am convinced that this would be harmful instead of helpful for the COSMIC data.

**References:**
Toohey, M., M. I. Hegglin, S. Tegtmeier, and the SPARC Data Initiative Team, Characterizing sampling bias in the trace gas climatologies of the SPARC Data Initiative, J. Geophys. Res., doi:10.1029/2013JD020298, 2013.
Ho, S., et al., Global Evaluation of Radiosonde Water Vapour Systematic Biases using GPS Radio Occultation from COSMIC and ECMWF Analysis, Remote Sens., 2, 1320-1330, doi: 10.3390/rs2051320, 2010.

---

## Referee Comment (RC2) · Anonymous Referee #2 · 1 Sep 2016

General comments The main subject is a validation of the water vapour content derived from COSMIC measurements. The COSMIC water vapour data quality is examined through comparisons with MLS, MERRA and ERA-Interim products. Water vapour variability in time and in space, on some selected pressure levels, has been investigated as well. The authors assume that the taken here approach, they call it "in a climate perspective", will allow to better estimate the quality of COSMIC water vapour through examinations in the upper troposphere and lower stratosphere layers.

Specific comments The compared data sets in the validation procedures are monthly and annually averaged secondary measurements. Such data averaging for comparison is broadly used in the research of atmosphere. It is a proper proper way if in the time

series of measurements there are only known periodical terms (like annual or monthly) and without secular trends. I afraid that it might be not the most optimal way here. In presence of some long periodical terms or some trend (secular changes), and if there are longer gaps in the analysed time series, the accuracy of statistical investigations will be degraded.

Unfortunately results of the comparisons are not clearly described, and the conclusions could be better justified.

Technical corrections Some technical corrections are also required. For example the wording "GPS-Radiosonde observations" is confusing, and must be changed. GPS receiver is used there not as a sounding sensor, like in RO missions, but only to trace 3D position of the sensors. Unfortunately I found more such examples in a number of publications on atmospheric measurements instruments. The sentence must be corrected. Also a portion of the first sentence on the page 4: "the time-frequency contents in the received signals" should be corrected/improved, clear formulated.

Unfortunately I have to share the opinion of the first reviewer that the paper "would be used and cited as justification for using COSMIC water vapour data in the stratosphere".

I recommend to improve, deeply modify the manuscript (consider all comments of the first reviewer) before publication.